

# The cuticle inward barrier in *Drosophila melanogaster* is shaped by mitochondrial and nuclear genotypes and a sex-specific effect of diet

Wei Dong[1,2], Ralph Dobler[2], Damian K. Dowling[3] and Bernard Moussian[1,2,4]

[1] Institute of Applied Biology, Shanxi University, Taiyuan, China
[2] Applied Zoology, Faculty of Biology, Technische Universität Dresden, Dresden, Germany
[3] School of Biological Sciences, Monash University, Clayton, Victoria, Australia
[4] Université Côte d'Azur, CNRS—Inserm, iBV, Parc Valrose, Nice, France

## ABSTRACT

An important role of the insect cuticle is to prevent wetting (i.e., permeation of water) and also to prevent penetration of potentially harmful substances. This barrier function mainly depends on the hydrophobic cuticle surface composed of lipids including cuticular hydrocarbons (CHCs). We investigated to what extent the cuticle inward barrier function depends on the genotype, comprising mitochondrial and nuclear genes in the fruit fly *Drosophila melanogaster,* and investigated the contribution of interactions between mitochondrial and nuclear genotypes (mito-nuclear interactions) on this function. In addition, we assessed the effects of nutrition and sex on the cuticle barrier function. Based on a dye penetration assay, we find that cuticle barrier function varies across three fly lines that were captured from geographically separated regions in three continents. Testing different combinations of mito-nuclear genotypes, we show that the inward barrier efficiency is modulated by the nuclear and mitochondrial genomes independently. We also find an interaction between diet and sex. Our findings provide new insights into the regulation of cuticle inward barrier function in nature.

Corresponding authors
Wei Dong, dongwei@sxu.edu.cn
Bernard Moussian,
bernard.moussian@unice.fr

## INTRODUCTION

The insect cuticle plays an important role in maintaining homeostasis by preventing uncontrolled penetration of xenobiotics and water (*Hadley, 1978*; *Lockey, 1976*; *Wang et al., 2016*). This barrier function relies mainly on the outer cuticular region composed of the envelope and surface lipids including cuticular hydrocarbons (CHCs) at the cuticle surface (*Blomquist & Bagnères, 2010*; *Gibbs, 1995*; *Gibbs, 2002*). In general, CHCs have chain lengths from C23 to C50, and may have double bonds and be branched. The CHC pool is species-specific and shows within-species variation with respect to age, sex and diet (*Barbosa et al., 2017*; *Bonelli et al., 2015*; *Ishii et al., 2002*; *Moore et al., 2017*; *Otte, Hilker & Geiselhardt, 2015*; *Rouault, Capy & Jallon, 2001*). Length variation of CHCs was found in different geographical populations of the fruit fly *Drosophila melanogaster*, where the ratio

of two CHC isomers varied with climatic conditions (*Ferveur, 1991*; *Ferveur & Sureau, 1996*). More recently, it was reported that the chain length of CHCs correlates with the latitudinal habitats of *D. melanogaster* strains from north (Maine) to south (Florida) along the US east coast paralleling increasing temperatures and desiccation threat (*Rajpurohit et al., 2017*). Rajpurohit and colleagues also found an association of genomic SNPs with the production and chemical profile of CHCs (*Rajpurohit et al., 2017*). Consistently, it has been shown that CHC variation in recombinant inbred lines depends on the nuclear genotype in *D. melanogaster* (*Dembeck et al., 2015*). Based on these works, it is conceivable that CHC chain length is associated with the function of the outward barrier and as a result would affect the desiccation resistance (*Rouault et al., 2004*).

Two findings indicate that also the inward barrier is based on lipids including CHCs. First, penetration of xenobiotics such as the dye Eosin Y is sensitive to lipid solvents (*Wang et al., 2016*; *Wang, Carballo & Moussian, 2017*). Second, cuticle impermeability for water and xenobiotics is disrupted by mutations in genes coding for proteins involved in lipid-based barrier formation (*Li et al., 2017*; *Yu et al., 2017*; *Zuber et al., 2018*). The molecular function of these proteins, including the ABCH transporter Snustorr (Snu) and the extracellular protein Snustorr-Snarlik (Snsl), and their relationship to CHC distribution, however, are yet unexplored. Hence, at least to some extent, the inward and outward barriers share the same molecular constitution.

Diet has a significant impact on lipid composition of *D. melanogaster* in general (*Brankatschk et al., 2018*; *Buczkowski et al., 2005*; *Carvalho et al., 2012*; *Liang & Silverman, 2000*; *Martin, Helanterä & Drijfhout, 2011*; *Savarit & Ferveur, 2002*; *Wurdack et al., 2015*). Food composition, sex and their interactions significantly affected CHC composition and amounts in strains of *D. melanogaster*, maintained on different food types for years (*Fedina et al., 2012*). Likewise, in the mustard leaf beetle *Pheadon cochleariae*, different food resources (host plants) contribute to variation in CHC profiles, which in turn affects their courtship behaviour (*Geiselhardt, Otte & Hilker, 2009*). The diversity of CHCs is assumed to depend on different metabolic pathways associated with digestion processes and adaptations to specific diets.

Mitochondria are important sites of lipid metabolism in the cell, thereby contributing to CHC production. Key steps of this process occur in these organelles. For instance, the bulk of acetyl-CoA required in fatty acid synthesis by the acetyl-CoA carboxylase is produced in the mitochondria (*Hardwood, 1988*; *Kennedy, 1962*; *Wakil, Stoop & Joshi, 1983*). Biosynthesis of haem, a co-factor of cytochrome P-450 enzymes, which modify the long carbon chains in the mitochondria or the smooth endoplasmic reticulum (ER) (*Capdevila, Falck & Estabrook, 1992*), involves enzymes acting in the mitochondria. While the role of mitochondria in lipid metabolism is well understood (*Goldin & Keith, 1968*; *Mesmin, 2016*; *Scharwey, Tatsuta & Langer, 2013*; *Tyurina et al., 2014*; *Voelker, 2004*), it is not known whether mitochondrial genetic variation is associated with variation of lipid or CHC synthesis, in turn influencing cuticle barrier function.

There are indeed some examples demonstrating the importance of mitochondrial genetic variation for insect ecology, especially in *D. melanogaster* (*Ballard, Pichaud & Fox, 2014*; *Camus & Dowling, 2018*; *Dowling, 2014*; *Wolff et al., 2014*). In the case of heat tolerance

in *D. melanogaster*, for example, *Camus et al. (2017)* recently showed that variation in this phenotype in Australia was in part associated with mitochondrial single nucleotide polymorphisms (SNPs) that did not change the protein sequence.

An issue complicating experimental approaches to assess the contribution by mitochondrial genetic variation on vital body functions is that mitochondrial effects (caused by variation in the mitochondrial genome) on the phenotype can differ across nuclear genomes, across sex and with environmental factors such as diet (*Aw et al., 2018*; *Zhu, Ingelmo & Rand, 2014*). It is well known that some organismal dysfunctions associated with mitochondrial mutations are expressed in only some nuclear backgrounds, but not in others (*Blumberg, Rice & Kundaje, 2017*; *Connallon et al., 2018*; *Dobler et al., 2014*; *Dobler et al., 2018*; *Dowling, 2014*; *Kenney et al., 2014*; *Latorre-Pellicer et al., 2016*; *Patel et al., 2016*; *Reinhardt, Dowling & Morrow, 2013*; *Wolff et al., 2014*). Several of these effects are sex-specific (e.g., *Immonen et al., 2016*). Addition of long-chain fatty acids to the diet has been shown to influence mitochondrial physiology (*Holmbeck & Rand, 2015*; *Stanley, Khairallah & Dabkowski, 2012*), and even offset mitochondrial genetic defects (*Senyilmaz et al., 2015*). Moreover, such nuclear or environmental influences on mitochondrial effects can differ across mitochondrial genotypes (*Ballard and Youngson, 2015*; *Mossman et al., 2016a*; *Mossman et al., 2016b*).

To detect an effect of genotypic mitochondrial variation on cuticle differentiation requires an advanced experimental protocol in which the effects of mitochondrial variation are isolated from diet, sex and nuclear genome effects. Here, we use a sophisticated experimental design employing *Drosophila* lines with distinct mito-nuclear genotypes reared on different diets. We coupled this with a new *in situ* method that generically measures the cuticle inward barrier function (*Wang et al., 2016*) and thereby examine the effects of mitochondrial and nuclear genomic variation on the cuticle inward barrier function.

This new *in situ* method is based on the ability of the dye Eosin Y to penetrate the cuticle, thereby reflecting the cuticle barrier function in *Drosophila* and other insects (*Wang et al., 2016*; *Wang, Carballo & Moussian, 2017*). Penetration of Eosin Y is regionalised, with different body parts taking up the dye at distinct temperatures. Eosin Y staining provides a simple and reliable way to detect cuticle inward barrier properties that are possibly mediated by lipids and CHCs. Here, we analyse the Eosin Y penetration pattern in the wing cuticle that allows fast and efficient assessment of differences in the cuticle inward barrier function caused by the mitochondrial genome, the nuclear genome, sex or diet.

## MATERIAL AND METHODS

### Fly line generation and maintenance

We created nine fly lines with different combinations of mitochondrial and nuclear genomes by specific crossing of males and females from three source populations. The three source populations originated from Coffs Harbour, Australia (**A**) (*Dowling, Williams & García-Gonzaléz, 2014*; *Williams et al., 2012*), Benin (**B**) (formerly Dahomey), Africa (*Clancy, 2008*) and Dundas (near Hamilton), Canada (**C**) (*MacLellan, Whitlock*

*& Rundle, 2009*). According to the "Climate Data for Cities Worldwide" database (https://en.climate-data.org/), Coffs Harbour lies in a zone with a humid subtropical climate (Cfa according to the climate classification of Köppen, average temperature of 18.8 °C, 1,688 mm of precipitation), Dundas lies in a warm-summer, humid continental climate (Dfb, average temperature of 8.5 °C, 834 mm of precipitation) zone and Benin lies in a zone with tropical monsoon climate (Aw, average temperature of 27.4 °C, 1,320 mm of precipitation). We initially crossed 45 virgin females from the source population with the desired mitochondrial genotype to 45 males from the source population with the desired nuclear genotype to create the first generation of each mito-nuclear line. To avoid skewed effects of mito-nuclear combinations due to non-random sampling from each source population, we created each line three times independently (resulting in a total of 27 lines) and kept them separated from each other since then. To generate the second generation (and all subsequent generations) we backcrossed 45 virgin female offspring from the line (harbouring the desired mitochondrial genotype due to maternal inheritance of the mitochondria *Birky, 2001*) with 45 males from the source population with the desired nuclear genotype (Fig. 1). With this crossing scheme, we removed 50% of the remaining nuclear genome from the maternal source population in each generation, leading to theoretical 99.99% removal of the maternal nuclear genome after 17 generations. We continued with the described crossing scheme for another 21 generations (until the experiment started) to avoid selective co-adaptation processes between the nuclear and the mitochondrial genome. We labelled the generated mito-nuclear lines as AA, AB, AC, BA, BB, BC, CA, CB and CC, the first letter denoting the origin of the mitochondrial genome and the second letter denoting the origin of the nuclear genome (hence mito-nuclear lines). We further distinguished the three replicates of each line with a suffix (1 to 3). We kept the mito-nuclear lines and the source populations as 14 day non-overlapping generations at 25 °C on a 12:12 h day-night rhythm. We kept these flies on 7 ml standard corn-yeast-sugar medium (corn 90 g/l, yeast 40 g/l, sugar 100 g/l, agar 12 g/l, Nipagin 20 ml/l, propionic acid 3 ml/l) in 25 mm vials.

All lines we used for the experiment were free of *Wolbachia*. This was confirmed for all lines by diagnostic PCR for *Wolbachia*- specific primers after infected lines were treated with Tetracycline (0.3 g/l added to the food) for three generations (*Clancy & Hoffmann, 2010*). We applied the Tetracycline treatment at least four generations before the start of our experiment.

## Food treatment

We used two distinct diets to assess the effect of food composition on the function of the cuticle barrier. The diets were developed by *Carvalho et al. (2012)* and *Brankatschk et al. (2018)* and differ in lipid composition while being isocaloric. One food type was plant food (PF, 788 kcal/l). Compared to the standard food, extra malt (80 g/l), cold pressed sunflower oil (2 ml/l) and treacle (22g/l) were added, while yeast and glucose were removed. The other food type was yeast food (YF, 809 kcal/l). Compared to the standard food, fresh yeast (80 g/l) and extra yeast extract (20 g/l) were supplied and cornmeal was removed. A detailed description of the recipes for the two food types can be found elsewhere

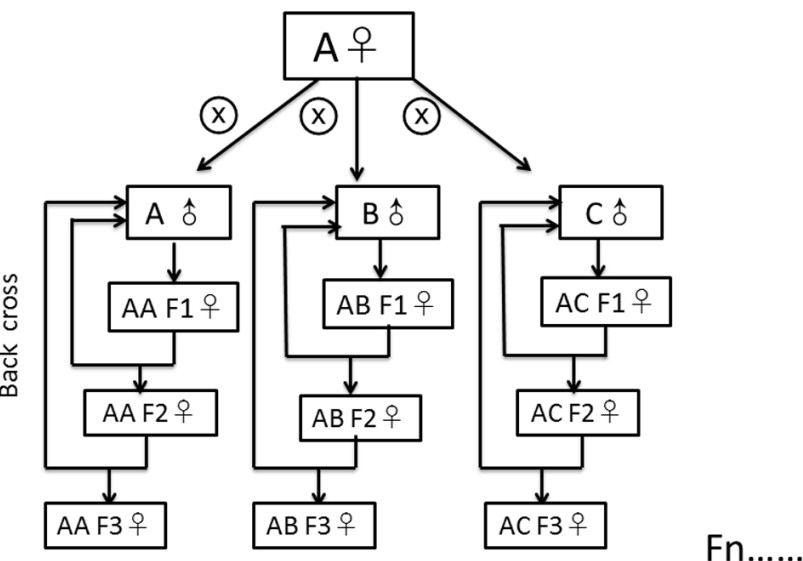

F1: the first generation   F2: the second generation   F3: the third generation

**Figure 1 Crossing scheme of fly line generation.** We initially crossed 45 virgin females from population (A) to 15 males from (A, B and C), respectively, to create the first generation. After then, we backcrossed 45 female offspring from the line with 45 males from the desired nuclear genotype. All subsequent generations repeat the back cross scheme. Similarly, we generated mito-nuclear lines as BA, BB, BC, CA, CB and CC by crossing males from the three source populations to females from the populations B and C, respectively.

(*Brankatschk et al., 2018*; *Carvalho et al., 2012*). In principle, the nutritional value of both types of food is similar. By contrast, their lipid composition differs: the plant food contains longer and more unsaturated lipids that the yeast food. The mito-nuclear lines completed an entire life cycle on either PF food or YF food (i.e., females laid eggs on the designated food type and larvae subsequently developed on this food type) before we collected adult flies for wing staining and measurements (see below). To roughly control larvae density, we used 10 males and 10 females to lay eggs and we standardised egg-laying time to 12 h for each line. Finally, we checked the egg density roughly by eye to make sure it was approximately equal across all lines. We kept all vials at 25 °C on a 12:12 h day-night rhythm. We collected virgin males and females within six hours of eclosion and kept them in vials (10 flies/vial) separated by sex and line for five days, thereafter we started wing staining and measurements.

## Wing staining and measurement

We used 10 flies (20 wings) of each combination (per line/food/sex) for wing staining. We carried out Eosin Y staining according to *Wang et al. (2016)* with a slight modification. Instead of two days old flies in the original protocol, we used five days old flies. We anaesthetized the flies with $CO_2$, transferred them into a micro-centrifuge tube containing

1 ml of the red dye solution (0,5% Eosin Y (W/V) and 0.1% Triton X-100) and incubated them at 55 °C for 30 min. We washed the Eosin Y-stained flies three times with distilled water, isolated wings using tweezers and mounted them in 50% glycerol on glass slides. We collected images using a Leica DMI8 microscope with a built-in camera and the software LAS X. For quantification of staining, we converted the images to 8-bit format and recorded the mean grey values that reflect the staining intensity using the Fiji software (*Schindelin et al., 2012*). We compared the mean grey values in the areas expected to take up Eosin Y and in surrounding areas. In a previous work, we discovered that the Eosin Y staining pattern depended on the genetic background of *D. melanogaster* (*Wang et al., 2016*). If the mean grey values of potentially Eosin Y-positive areas (in the posterior, lower half), were higher than those of potentially Eosin Y-negative areas (in the anterior, upper half), we scored a "presence of staining", otherwise we scored an "absence of staining". We used a semi-quantitative method to classify staining patterns as 'no staining', 'front area staining', 'back area staining' or 'front and back area staining'.

## Statistical analyses

Prior to data analysis we inspected our data and found that one staining pattern (front patch unstained, back patch stained) occurred in only seven out of 1108 individuals distributed across four of the nine mito-nuclear combinations. We omitted these seven measurements from our analyses and used the remaining 1101 measurements assigned to one of the three staining pattern as ordinate response variable (0, 1, 2 stained spots) for our analysis. We analysed the data with generalised linear mixed models (GLMMs) using the *lme4* package (*Bates et al., 2013*) in R 3.4.2. (*R Development Core Team, 2017*). We started with a full model including the factors diet, sex, mitochondrial and nuclear genome and all their higher-order interactions. We then reduced the model stepwise by excluding factor combinations to improve the Bayesian information criterion (BIC). We used the BIC for model comparison because of the large number of degrees of freedom in our models. Using BIC reduces the chance to have false positive factors in the final model because a high number of degrees of freedom is more penalised than with the AIC (*Dziak et al., 2012*). We stopped the model reduction when the removal of factor combinations did not increase the explanatory power of the model (the final model). The four linear factors remained in the final model because they were experimentally manipulated and we were *a priori* interested in their effects.

The full statistical model included the observed patterns as dependent variable (ordinal data type). Mitochondrial genotype, nuclear genotype, sex and food type, with all their higher-level interactions, were fixed effects of the full model and line was used as random factor to avoid pseudo-replication of data in the analyses. As the response variable was ordinal (see above), we used a Binomial distribution with a log-link error function to analyse the data. We further changed the number of maximal iterations for the model to converge from 1,000 to 500,000 in the glmer Control to assure model convergence. To find the optimal grouping of the fly lines we ran a principal component analysis (PCA) using SNP analyses of the mitochondrial genomes from all our mito-nuclear lines (GenBank accession PRJNA532313) and the origin of the nuclear genome. In brief we ran PCAs on

**Table 1  ANOVA table for the statistical analysis of the wing pattern frequency.** Analysis of deviance table (type III Wald $\chi^2$ test) for the optimised model. The interaction between sex and food was statistically significant, with both of the two linear terms to be found non-significant. The mitochondrial genotype and the nuclear genotype had also a significant effect on the staining pattern.

| Factor | $\chi^2$ | df | p-value |
|---|---|---|---|
| Intercept | 21.337 | 1 | <0.001 |
| Mitochondrial genotype | 12.740 | 2 | 0.002 |
| Nuclear genotype | 134.358 | 2 | <0.001 |
| Sex | 0.161 | 1 | 0.688 |
| Food | 0.646 | 1 | 0.422 |
| Sex × food | 14.146 | 1 | <0.001 |

the frequencies of the three observed staining patterns using 34 SNPs and the nuclear background as explanatory variables. The aim of the PCAs was to see whether and how the 27 lines cluster in the plane of the first two principal components. Based on the results from the PCA we grouped the fixed factor mitochondrial genotype to three levels (Fig. S1). We grouped the mitochondrial genotypes AA1, AA2, AB1-3 and AC1-3 (hereafter type 'A'), the mitochondrial genotypes BA1-3, BB1-3, BC1-3, CA1 and CA3 (hereafter type 'B') and the mitochondrial genotypes AA3, CA2, CB1-3 and CC1-3 (hereafter type 'C').

## RESULTS

We scored the frequency of occurrence of two Eosin Y staining areas at the posterior half of the wing blade of flies from population lines with putatively co-evolved or newly constituted mitochondrial and nuclear genome combinations in order to test barrier efficiency. It should be noted that as these fly lines were derived from natural populations with assumed nucleotide diversity in the mitochondrial and nuclear genomes, and not from isogenic stocks, we did not expect to observe only one staining pattern in flies from a single population. We found variation in the wing-staining pattern across our *D. melanogaster* lines (Fig. 2). The frequencies of the wing staining patterns for each of the four fixed factors are visualised in Fig. S2.

The final model revealed that nuclear ($p < 0.001$, Table 1) and mitochondrial ($p = 0.002$) genotypes, as well as the interaction between sex and food ($p < 0.001$), best explained the frequency of the wing-staining pattern (Table 1, Figs. 3 and 4). Of note, our data do not provide evidence for a significant mito-nuclear interaction effect on the inward barrier function of the cuticle ($p = 0.141$).

For different nuclear genotypes, flies with the Australian nuclear genotype generally showed the highest frequency of non-stained wing area (39.94%) and the lowest frequency of both wing areas being stained (43.30%) among all three nuclear genotypes (Fig. 4 and Fig. S1). Flies with both Benin and Canadian nuclear genotypes showed high frequencies of double stained areas patterns (Benin: 96.26%, Canada: 74.80%) (Fig. 4 and Fig. S1). The frequency of wing staining pattern with only the rear area being stained was about the same in flies with the Australian and the Canadian nuclear genotype (Australia: 16.76%;

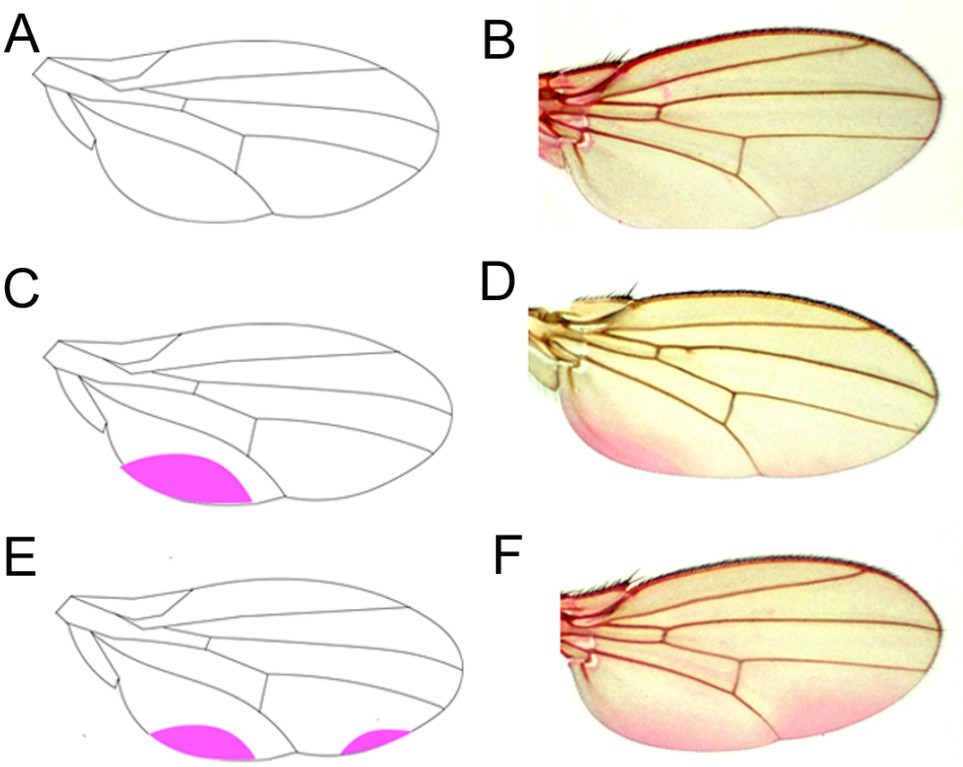

**Figure 2 Eosin Y staining pattern of the fly wing.** As represented by a drawing (A), a proportion of wings does not take up Eosin Y after staining (B). Some wings, by contrast, show staining of a posterior region close to the hinge (C, D). Another set of wings takes up the dye in two posterior regions (E, F).

Canada: 17.07%) (Fig. 4 and Fig. S1). This pattern was almost absent in flies with the Benin nuclear genotype (1.07%) (Fig. 4 and Fig. S1).

For different mitochondrial genotypes, the staining frequency for the rear area was similar in all three mitochondrial genotype groups (A: 10.80%, B: 14.96%, C: 7.60) (Fig. 4), whereas there was substantial variation in whether no or both wing spots were stained (no staining: A: 19.44%, B: 18.08%, C: 11.85%; staining both areas: A: 69.75%, B: 66.96%, C: 80.55%) (Fig. 4).

The significant interaction effect between food and sex is underlined by the striking differences of a male–female difference in wing staining pattern for plant food or yeast food (Fig. 4). Specifically, most wings of flies reared on plant food showed staining at both posterior areas (female: 234, male: 252), and only a few wings of these flies had non-stained wings (female: 29, male: 34). The frequency of wing staining patterns with only the rear area being stained was lowest in males reared on plant food (Table S1, Fig. S2).

## DISCUSSION

The physiological and genetic mechanisms involved in the constitution of the inward barrier function of the insect cuticle are largely unexplored. To contribute to our understanding

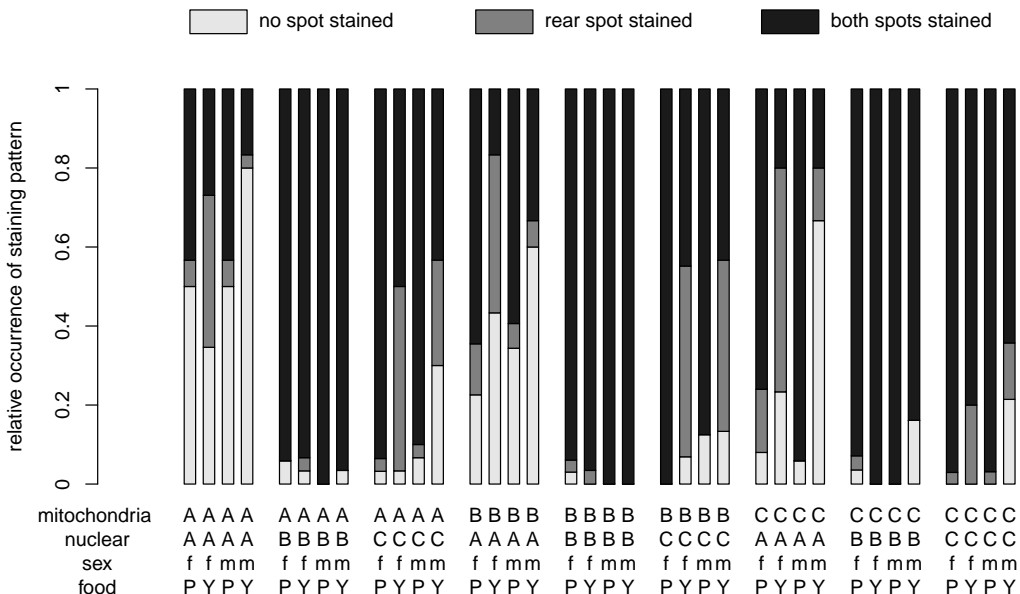

**Figure 3** **Relative frequencies of wing patterns.** The relative occurrence of each wing pattern for each mitochondrial genotype - nuclear genotype -sex-food combination. A, B and C in the first line of the x-axis label are for the mitochondrial genotype. The A, B and C in the second line stand for the nuclear genotype. The third line indicates the sex of the flies m for males and f for females and the forth line label are for plant food (P) or yeast food (Y).

of these mechanisms, we investigated the effects of nuclear and mitochondrial genetic variation, sex and diet and their interactions on the function of the wing cuticle inward barrier. These factors had an effect on cuticle barrier function at different hierarchical levels.

## Interactions between sex and diet

According to our statistical model, if we consider sex or diet alone (as main effects), they do not have any significant effect on the frequency of the wing-staining pattern and, by consequence, on the cuticle inward barrier efficiency. Yet, they are involved in a statistically significant interaction, with the effects of each of these factors is contingent on the other. Simply, this suggests an important role of sex on cuticle barrier function that is modulated by diet. Thus, with respect to the cuticle barrier function, we conclude that the responses to food quality between males and females are different. In other words, food utilization for barrier construction and optimization (that may at least partially rely on CHCs) varies between males and females. Indeed, in recent studies it was shown that the impact of the food source on the CHC pattern differed in males and females (*Otte, Hilker & Geiselhardt, 2015*). Thus, it seems that the CHC composition is not only sexually dimorphic with respect to communication (*Antony & Jallon, 1982*), but also with respect to penetration resistance against xenobiotics.

**Frequencies of wing patterns**

**Figure 4** **Nuclear, mitochondrial and interaction of sex and diet on wing pattern frequencies.** The relative occurrence of each wing pattern for each mitochondrial genotype, nuclear genotype and sex- food combination, respectively. The interaction between sex and diet affects the number of wings with no staining and with a single spot staining, but not the number of wings with both spots stained. Females (f) on yeast food (Y) show a much higher proportion of single stained wings than males (m) on yeast food and males and females on plant food (P). Males on yeast food have a higher occurrence of unstained wings than females on yeast food and males and females on plant food. In summary, males and females show similar frequencies for double stained area (m: 71.61%, f: 72.09%). However, the frequencies between non-stained (m: 21.61%, f: 11.46%) and single stained areas (m: 6.79%, f: 16.45%) showed some difference. Staining patterns between plant food and yeast food were quite different (non-stained: P: 11.07%, Y: 22.56%; single stained: P: 3.51%, Y: 20.11%; double stained: P: 85.41%, Y: 57.33%).

Next, we need to determine sex-specific single nucleotide polymorphisms (SNPs) and identify the CHC composition in males and females of our lines on different food sources to better understand inward barrier function in dependence of diet and sex. These experiments would also allow us to reveal possible differences and similarities between the molecular constitution of the inward and the outward barrier for which rich data is available.

Indeed, a difference of the outward barrier between sexes has been observed in *D. melanogaster*. Due to specific modifications of CHC composition after induction of desiccation resistance, the survival rate of females is higher than the survival rate of males under prolonged desiccation conditions (*Foley & Telonis-Scott, 2011*; *Stinziano et al., 2015*). This effect is independent of starvation. These findings with respect to our results are interesting insofar as they suggest that the outward and inward barriers display independent efficiencies; while, for instance, females are more resistant to water loss than males, resistance to dye penetration is comparable between females and males if we ignore the effect of diet (see below). A simple conclusion is that these two barriers do not rely on

CHCs and the respective physical properties alone, but employ distinct and non-analogous factors. Further experiments are needed to unravel the molecular basis of this difference.

Likewise, it has previously been demonstrated that diet does control the efficiency of the outward barrier, probably by influencing the amounts and, importantly, the composition of CHCs (*Fedina et al., 2012*). In line with this work, a recent study reported that the proportion of desaturated CHCs defines the efficiency of the barrier against desiccation (*Ferveur et al., 2018*). This may be explained by an influence of food metabolites on fatty acid synthesis and, by consequence, on CHC amounts and composition (*Fedina et al., 2012*; *Ferveur, 2005*; *Pavković-Lučić et al., 2016*). However, the situation is certainly more complex. The effect of food alone on CHC quality is not sufficient to explain the outward barrier function. Indeed, the effects of food source on desiccation resistance have been repeatedly reported to depend on multiple complex factors including, in addition to lipid metabolism, carbohydrate metabolism and body size (*Andersen et al., 2010*; *Kristensen et al., 2016*; Mikkelsen et al., 2010). Moreover, diet has been demonstrated in large-scale studies to have a significant and variable effect on genome-wide gene expression in 20 *D. melanogaster* wild-type strains in various complex traits including puparial adhesion, metamorphosis and central energy metabolic functions (*Reed et al., 2010*; *Reed et al., 2014*; *Williams et al., 2015*). Overall, we therefore conclude that the outward and inward barriers are differentially sensitive to diet.

## The nuclear genotype

We observed significant effects of the nuclear genotype on the dye penetration efficiency. In particular, the nuclear genotype of flies from Coffs Harbour with a humid subtropical climate (Cfa according to the climate classification of Köppen) correlates with a wing cuticle inward barrier that is more efficient than in flies with the Dundas (warm-summer, humid continental climate, Dfb) or with the Benin (tropical monsoon climate, Am) nuclear genotype, which correlates with the lowest cuticle inward barrier efficiency. Rough temperature or humidity profiles of the locations (see Materials & Methods) do not explain these correlations arguing that the cuticle inward barrier may not be directly dependent on these factors. In any case, we conclude that the inward barrier function depends largely on varying fly line-specific expression profiles of nuclear genes, some of which presumably are needed for CHC production and deposition including genes coding for Cyp proteins, fatty acid synthases and elongases. Nuclear genetic variations that comply with differences in CHC composition in inbred or geographically separated lines of *D. melanogaster* have been reported to be represented by quantitative trait loci (*Foley et al., 2007*; *Foley & Telonis-Scott, 2011*) or SNPs (*Dembeck et al., 2015*; *Rajpurohit et al., 2017*). The genomic differences between our lines and the associated differences in cuticle permeability are either due to genetic adaptation to the original environmental conditions in Coffs Harbour, Australia, Benin, Africa and Dundas (near Hamilton), Canada, or due to genetic drift within the populations. This remains to be tested.

## The mitochondrial genotype

We found a significant effect of the mitochondrial genotype on the wing-staining frequency patterns. Cuticle inward barrier efficiency was similarly reduced in flies with

a Dundas mitochondrial genotype. Rough climate profiles of the original locations of our fly populations are insufficient to explain the correlation between the mitochondrial genotype and cuticle inward barrier efficiency. More detailed assessment of the local climate situation and the use of more geographically disjunct fly populations are thus necessary to allow relating cuticle inward barrier efficiency to any climatic factor. A relationship between cuticle inward barrier efficiency, mitochondrial genotype and a climatic factor would, in turn, allow the testing of whether this relationship is in agreement with the *mitochondrial climatic adaptation* hypothesis (*Camus et al., 2017*). This hypothesis proposes that latitudinal climatic differences shape patterns of standing variation in mitochondrial genotypes across a species distribution, and that these genotypes play a role in determining temperature sensitivity of individuals.

In a recent work, no significant effect of the mitochondrial genotype on lipid content was found in *D. melanogaster* lines engineered from laboratory and natural populations (*Aw et al., 2017*). That study thus indicated that lipid (including CHC) homeostasis is to a large extent independent from the mitochondrial genotype in *D. melanogaster*. Together, we conclude that the function of the inward barrier is sensitive to variation in the mitochondrial genomes of the lines tested. We also conclude that the mitochondrial-dependent inward barrier function is possibly not mediated by CHCs.

Interestingly, Aw and colleagues (*2017*, *2018*) did find a significant interaction of the mitochondrial genotype with diet and/or sex on a number of physiological traits including survival and fecundity. According to our data, however, these genotype-by-environment interactions do not seem to play any role with respect to the cuticle inward barrier function. Likewise, we do not detect any interaction between the mitochondrial and the nuclear genomes that would influence dye penetration in our assays. The seeming independence of the inward barrier function from mito-nuclear interaction possibly represents a particular case. Indeed, mito-nuclear interactions have been demonstrated to affect a wide range of biological processes such as developmental time, sex-specific transcription, hypoxia and longevity (*Dowling, Meerupati & Arnqvist, 2010*; *Mossman et al., 2016a*; *Mossman et al., 2016b*; *Mossman et al., 2017*; *Rand, Fry & Sheldahl, 2006*; *Rand et al., 2018*). We will need to expand the number of different geographical populations of *D. melanogaster* or refine the quantification of our dye-penetration assay in our analyses to uncover whether any subtle mito-nuclear effects on cuticle barrier function exist.

## CONCLUSIONS

In summary, in this study we find that Eosin Y penetration through the wing cuticle of mass bred populations from three global populations is variable and that the pattern of variability differs between the lines. Our data indicate that along with CHCs, other cuticle components are sensitive to genetic variation within the nuclear and mitochondrial genomes as well as to sex-diet interactions. Candidates are factors acting in the Snu-Snsl pathway that contribute to the construction of the outermost cuticle layer termed envelope that serves as a physical barrier against penetration and desiccation (*Zuber et al., 2018*). Nuclear and mitochondrial SNP analyses will help to shed light on this complex trait

that is crucial for insect survival. Mitochondrial genes comprising 13 polypeptides of the electron transfer-chain (ETC), 2 rRNAs and 22 tRNAs, are, however, probably not directly involved in cuticle barrier formation (*Ballard & Rand, 2005*; *Burton, Pereira & Barreto, 2013*; *Piomboni et al., 2012*; *St John, Jokhi & Barratt, 2005*).

## ACKNOWLEDGEMENTS

We thank Klaus Reinhardt and three anonymous referees for helpful comments on previous versions of the manuscript.

### Funding

The study was financially supported by the Deutsche Forschungsgemeinschaft-Exzellenzinitiative Zukunftskonzept to Technische Universität Dresden (Ralph Dobler, and Bernard Moussian), by the Australian Research Council: DP170100165 and FT160100022 (to Damian K. Dowling), by National Natural Science Foundation of China (NSFC31761133021,31402021), Special Talents Projects in Shanxi Province, China [201805D211019] (to Wei Dong) and the Deutsche Forschungsgemeinschaft [DFG1714/9-1] (to Bernard Moussian). The funders had no role in study design, data collection and analysis, decision to publish, or preparation of the manuscript.

### Grant Disclosures

The following grant information was disclosed by the authors:
Deutsche Forschungsgemeinschaft-Exzellenzinitiative Zukunftskonzept to Technische Universität Dresden.
Australian Research Council: DP170100165, FT160100022.
National Natural Science Foundation of China: NSFC31761133021, 31402021.
Special Talents Projects in Shanxi Province, China: 201805D211019.
Deutsche Forschungsgemeinschaft: DFG1714/9-1.

### Competing Interests

The authors declare there are no competing interests.

### Author Contributions

- Wei Dong and Ralph Dobler conceived and designed the experiments, performed the experiments, analyzed the data, contributed reagents/materials/analysis tools, prepared figures and/or tables, authored or reviewed drafts of the paper, approved the final draft.
- Damian K Dowling analyzed the data, contributed reagents/materials/analysis tools, authored or reviewed drafts of the paper, approved the final draft.
- Bernard Moussian conceived and designed the experiments, analyzed the data, contributed reagents/materials/analysis tools, prepared figures and/or tables, authored or reviewed drafts of the paper, approved the final draft.

## Data Availability

The raw statistics are available in the Supplemental Files. The SNP data is available at GenBank: PRJNA532313.

## Supplemental Information

Supplemental information for this article can be found online at http://dx.doi.org/10.7717/peerj.7802#supplemental-information.

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
