# Peer review of "The cuticle inward barrier in Drosophila melanogaster is shaped by mitochondrial and nuclear genotypes and a sex-specific effect of diet"

_PeerJ, doi:10.7717/peerj.7802_

## Round 0.1 · original submission · Major Revisions

Dear Dr. Dong and colleagues:

Thanks for submitting your manuscript to PeerJ. I have now received three independent reviews of your work, and as you will see, the reviewers raised some concerns about the research. Despite this, these reviewers are optimistic about your work and the potential impact it will have on research communities studying fruit fly biology, physiology and evolution. Thus, I encourage you to revise your manuscript, accordingly, taking into account all of the concerns raised by both reviewers.

There are many suggestions by the three reviewers that will greatly improve your manuscript (mostly grammatical or minor content/clarity). While the concerns of the reviewers are relatively minor, this is a major revision to ensure that the original reviewers have a chance to evaluate your responses to their concerns.

I look forward to seeing your revision, and thanks again for submitting your work to PeerJ.

Good luck with your revision,

-joe

Reviewer 1 ·

Basic reporting

The manuscript is well written, with introductory sections conveying to the main points. The literature is generally relevant, and I provide some additional suggestions to make it even more accurate. The Figures could be improved (see below).

Experimental design

The experimental design is clear and the research questions well defined. The methods have been described with enough details.

Validity of the findings

The findings are well supported by the data based on a consistent number of replicates, and the statistics, to my level of expertise, are correct. While the discussion is clear, it could be still a little bit more speculative given the few research reports in this specific field of investigation. The conclusion are well stated and linked to the original questions and limited to supporting results.

Additional comments

In this paper, Dong and collegues explored the relationship between mitochondrial/nuclear genotypes and the permeability of insect cuticle. Additionally, the authors also explored the interaction between the sexes and two types of diet (plant or yeast based) in Drosophila melanogaster. They compared three populations from three geographical distant origins and used repeated backrosses to each male strain to keep constant the mitochondrial DNA from the ancestral female line and progressively introgress most of the nuclear DNA from the male population.

This very fundamental and original question pertaining to insect cuticle has received very little attention probably due to the limitation of tools. The authors designed an original trick, based on the use of a red vital dye (EosinY), to visualize and compare cuticule permeability in Drosophila wings. This simple, yet efficient trick, allowed them to semi-quantify the dye penetration into different wing areas, under controled conditions. To my point of view, this paper is an original and solid contribution which should be used in the near future as a landmark by entomologists and toxicologists.

My general impression after reading the paper is very positive and I have only few minor comments to improve the readability of the paper. The manuscript is well written, with introductory sections conveying to the main points. The literature is generally relevant, and I provide some additional suggestions to make it even more accurate. The Figures could be improved (see below). The findings are well supported by the data based on a consistent number of replicates, and the statistics, to my level of expertise, are correct. The experimental design is clear and the research questions well defined. The methods have been described with enough details. While the discussion is clear, it could be still a little bit more speculative given the few research reports in this specific field of investigation. The conclusion are well stated and linked to the original questions and limited to supporting results.

Minor comments:
L 118: is the average temperature in Benin 8.5°? It seems to be very low.
L129: I would write “...theoritical 99.99% removal..” since some regions of the chromosome are resistant to crossing over.
L180: you shoud add: “We used a semi-quantitative method to classify staining patterns as...”
LL 220-221: Why did the authors not expect to observe only one staining pattern? Explain.
L224: Table 1.
L263: The reference provided is uncorrect and should be replaced by “Antony and Jallon 1982 J.Insect Physiol” which is the first research paper showing sexual dimorphism in D.melanogaster cuticular pheromones.
LL 282-292: In this paragraph, I suggest you cite a recent PeerJ paper (Ferveur, JF., Cortot, J., Rihani, K., Cobb, M. Everaerts, C. (2018). Desiccation resistance : effect of cuticular hydrocarbons and water content in Drosophila melanogaster adults. PeerJ. 6: e4318.) which explored different parameters related to desiccation resistance.
LL305-307: Could the authors provide a more precise information about the nuclear genes (or families of genes) which may be potentially involved?
LL319-320/LL 343-345: I agree with the authors that they must be cautious about their conclusions based on the small number of populations sampled.

Figures: I had several problems with the Figures.
1- All figures should be oriented in “horizon” format” so you do not need to twist your neck to read them.
2- Figure 3 is impossible to read given that it contains too many information compacted in a unfriendly series of graphs. The authors shoud find a more gentle way to show/share their data. Either they choose different colors to distinguish between diet, sexes, genotypes etc.. or they introduce a gap between groups of bars based on their categories. The legends below the bars should also be shown in a more didactic way, maybe on two rows, one for the genotype and the second one with the sex/food regime. Currently it is quite impossible to read them.

Reviewer 2 ·

Basic reporting

Overall the manuscript is clear and well written. Below are minor comments on the text and the English.

1- Lines 116-118. The Köppen-Geiger climate classification for Dundas and Benin appears to be inverted (the average temperature for Benin is not 8.5oC but closer to 26oC, etc).
2- Line 190. Shouldn’t it say “did NOT improve”?
3- There is a number of sentences with awkward or incorrect English that should be revised:
- Lines 23, 175: rather than “unravel” I would use “determine”
- Line 156-157 “To approximately control larvae density, we...”. I would recommend using “roughly”
- Line 205 should say “principal” not “principle”
- Sentence line 265-268 is awkward.
- Sentence starting at line 271 should directly follow sentence that ends line 270 (since they are logically linked by “Indeed...”)
- Lines 275-276 Should say “... results are interesting insofar as they suggest that...”
- Line 282 Should say “...it has previously been demonstrated that diet does...”
- Sentence line 284-286 is awkward.
- Line 288 I would suggest using “affect” instead of “concern”
- Line 289: “along with” should be deleted (and comma moved to the left of “including”)
- Line 294, commas after “we” and “therefore” are not needed.
- Line 311 Should say “.... cuticle permeability are either due to genetic adaptation...”
- Line 316, I would suggest using “similarly” instead of “comparably”
- Line 322 Should say “...would, in turn, allow the testing of whether...”
- Line 324 Should say “This hypothesis proposes that latitudinal...”
- Paragraph lines 356-358 should follow line ending line 355. As a free-standing paragraph is appears to “come out of nowhere”
- Line 729: I would suggest deleting “(take AA, AB, AC for example)” from legend
- Sentence lines 751-752 is not clear/difficult to read.
- Legend to Table S1 (starting Line 763) has some unnecessary information since it is obvious when reading the table: “The left half shows the numbers for males, the right half the numbers for females.” And “The top half shows numbers for flies on plant food, the lower half the numbers for flies on yeast food.” I would also label the columns “Nuclear genotypes” (to the left of the A, B, C row) and rows “Mitochondrial genotype” (above the A, B or C column), and eliminate explanatory text from legend.
- Line 779, please link “We distinguish three groups.” to the rest of the legend; as it stands it appears disjointed.

Experimental design

I am no expert in the statistical methods used for the analyses, so cannot comment on the appropriateness of the criteria and tests used. But the rest of the work is well done and documented, and addresses an important question in insect biology.

Validity of the findings

I am no expert in the statistical methods used for the analyses, so cannot comment on the appropriateness of the criteria and tests used. But the rest of the work is well done and documented, and addresses an important question in insect biology.

Additional comments

This manuscript reports the broad pattern of inheritance of factors that regulate cuticle inward barrier function in Drosophila. Through a careful crossing scheme between populations from 3 different climates the authors were able to isolate the contribution of nuclear and mitochondrial genotype, and determine the role of sex and nutrition and the interaction between these factors, in shaping the insect’s ability to repel dye penetration through its cuticle.

My only apprehension is that the work is quite preliminary: it would be nice to know the identity of some of the factors that are important regulators of cuticle inward barrier function. For instance, directly measuring the levels of the different classes of cuticular hydrocarbons (CH) and their pattern of inheritance might go a long way towards understanding how this barrier varies amongst different populations. Likewise, a number of possible candidate mitochondrial genes are mentioned (lines 63-71), and their level of expression could be measured in the various sublines. The expression of candidate nuclear genes could also be determined, including genes relevant to CH synthesis, cuticular proteins, etc. I recognize that these experiments might involve a significant amount of work. The current report is (to my mind) complete as it is but the results of the suggested experiments would qualitatively improve the relevance and impact of this research.

Reviewer 3 ·

Basic reporting

no comment

Experimental design

no comment

Validity of the findings

no comment

Additional comments

The manuscript by Dong and colleagues reports a study on the variations of cuticle inward barrier among Drosophila strains. The insect cuticle is mediating most interactions with the external milieu and this complex structure has certainly contributed to the spectacular evolutionary success of insects. Moussian is a renowned specialist of the cuticle and his work has provided many insights into genes and regulatory mechanisms underlying cuticle formation during Drosophila development. In this manuscript, the authors focus on the inward barrier properties of the Drosophila adult cuticle and the impact of genotypes and other factors. Cuticle composition regulates the penetration of external compounds and water, the latter being particularly important for the adaptation of Drosophila populations to varying ecological conditions, such as temperature and humidity. To address these issues, the authors used Drosophila strains from three source populations (Australia (A), Benin (B) and Canada (C)). The general strategy is based on a simple staining assay of adult wings, using patterns of Eosin Y penetration as a readout of cuticle permeability. Quantification was done by semi-automated image analysis, as described in the corresponding section, and staining patterns were classified in 4 discrete categories. Using this approach, the authors detected clear differences in Eosin staining patterns between A, B and C strains, without obvious correlation with climate classification. Since cuticle composition may relies on other parameters, such as mitochondria (that synthesize lipid-related metabolites), food and sex, the authors used a smart crossing scheme to exchange mitochondrial genotypes between strains, as well as a relevant sampling design to assay for any possible interactions (nuclear, mitochondria, sex and food). The main results of this studies are the influence of both nuclear and mitochondrial genotypes, and a sex/diet interaction, on the inward barrier efficiency.

The manuscript is well written, the material and method section is detailed, and interpretations seem appropriate to describe and discuss the data.
As detailed below, I only have some suggestions to improve the manuscript.


Main points
1) Since the whole paper relies on a “new in situ method … based on the ability of the dye Eosin Y to penetrate the cuticle” (l99), some experimental validation of this assay and of its relevance to cuticle barrier would strongly strengthen the study. For example, Eosin patterns in a wild type strain versus a mutant background known to affect the barrier function should be very informative.

2) As suggested by the PCA chart (Fig S2), the stronger effect observed is the nuclear genotype, as also supported by fig4. It should be important to highlight these findings, both in the text and also by reordering the discussion section.

Minor points
3) I’m very surprised by climatic features of Benin (l118) and suspect some inversion between B and C. Also, I guess that the climate classification of Köppen is “Aw” instead of “Am”. The authors should carefully double check these points and correct them accordingly.

4) The formulation “lower half at 55°C” or “upper half at 55°C” (l177-78) remains unclear for me. This should be rephrased.

5) Also, the sentence “high frequencies of double stained areas patterns (female: 234, male: 252) and low frequencies of non-stained (female: 29, male: 34).”(l242-43) should be rephrased since it seems to refer to effectives instead of frequencies?

6) The notion of “statistically significant” (l255) is vague. There is a vivid debate on its use and misuse in biology (eg PMID 30894741), and whether it was ever useful… The American Statistical Society is also advocating for an end to statistical significance and urge more thoughtfulness in interpreting data (see the special issue of The American Statistician (2019), 73).

---

## Round 0.2 · Minor Revisions

Dear Dr. Dong and colleagues:

Thanks for revising your manuscript. The reviewers are mostly satisfied with your revision (as am I). Great! However, per reviewer 2, there are a few minor issues to address. Please address these issues ASAP so we may move towards acceptance of your work.

-joe

Reviewer 1 ·

Basic reporting

The authors have taken into consideration all my requirements, and therefore I think the paper can be accepted as it is.

Experimental design

valid (as in the first version)

Validity of the findings

valid (as in the first version)

Additional comments

The authors have taken into consideration all my requirements, and therefore I think the paper can be accepted as it is.

Reviewer 2 ·

Basic reporting

No comment.

Experimental design

No comment.

Validity of the findings

No comment.

Additional comments

The authors have revised the manuscript to my satisfaction. In particular the figures are much clearer and "reader friendly".
I only have a few suggestions for sentences that seem awkward to me:

Lines 23 “Testing different combinations of mito-nuclear genotypes, we unravel that the inward barrier efficiency...” Rather than “unravel” I would suggest using “show”.

Line 45 “...outward barrier and by consequence the desiccation resistance...” Rather than “by consequence” I would suggest using “as a result would affect”

Line 193 “... model because a high number of degrees of freedom is more penalised as with the AIC...” This sentence is not clear. Should it not say “...is more penalised THAN with the AIC...”?

Line 209 Sentence should start with “The” (“The aim...”).

Line 216 “We scored for the frequency of two Eosin Y staining...” I would suggest writing “We scored the frequency of occurrence of two Eosin Y staining...” (delete “for” and add “occurrence of”)

Reviewer 3 ·

Basic reporting

no comment

Experimental design

no comment

Validity of the findings

no comment

Additional comments

The authors addressed the points raised and their paper is ready for publication

---

## Round 0.3 · accepted · Accept

Dear Dr. Dong and colleagues:

Thanks for re-submitting your revised manuscript to PeerJ, and for addressing the concerns raised by the reviewers. I now believe that your manuscript is suitable for publication. Congratulations! I look forward to seeing this work in print, and I anticipate it being an important resource for research communities studying fruit fly biology, physiology and evolution..

Thanks again for choosing PeerJ to publish such important work.

-joe